# Synthetic Arterial Spin Labeling MRI of the Kidneys for Evaluation of Data Processing Pipeline

**DOI:** 10.3390/diagnostics12081854

**Published:** 2022-07-31

**Authors:** Irène Brumer, Dominik F. Bauer, Lothar R. Schad, Frank G. Zöllner

**Affiliations:** Computer Assisted Clinical Medicine, Mannheim Institute for Intelligent Systems in Medicine, Medical Faculty Mannheim, Heidelberg University, 68167 Mannheim, Germany; dominik.bauer@medma.uni-heidelberg.de (D.F.B.); lothar.schad@medma.uni-heidelberg.de (L.R.S.); frank.zoellner@medma.uni-heidelberg.de (F.G.Z.)

**Keywords:** renal perfusion, arterial spin labeling, synthetic data, magnetic resonance imaging, kidney

## Abstract

Accurate quantification of perfusion is crucial for diagnosis and monitoring of kidney function. Arterial spin labeling (ASL), a completely non-invasive magnetic resonance imaging technique, is a promising method for this application. However, differences in acquisition (e.g., ASL parameters, readout) and processing (e.g., registration, segmentation) between studies impede the comparison of results. To alleviate challenges arising solely from differences in processing pipelines, synthetic data are of great value. In this work, synthetic renal ASL data were generated using body models from the XCAT phantom and perfusion was added using the general kinetic model. Our in-house developed processing pipeline was then evaluated in terms of registration, quantification, and segmentation using the synthetic data. Registration performance was evaluated qualitatively with line profiles and quantitatively with mean structural similarity index measures (MSSIMs). Perfusion values obtained from the pipeline were compared to the values assumed when generating the synthetic data. Segmentation masks obtained by semi-automated procedure of the processing pipeline were compared to the original XCAT organ masks using the Dice index. Overall, the pipeline evaluation yielded good results. After registration, line profiles were smoother and, on average, MSSIMs increased by 25%. Mean perfusion values for cortex and medulla were close to the assumed perfusion of 250 mL/100 g/min and 50 mL/100 g/min, respectively. Dice indices ranged 0.80–0.93, 0.78–0.89, and 0.64–0.84 for whole kidney, cortex, and medulla, respectively. The generation of synthetic ASL data allows flexible choice of parameters and the generated data are well suited for evaluation of processing pipelines.

## 1. Introduction

Kidneys are mainly responsible for the excretion of waste materials produced by body metabolism via urine [1]. They are also key for maintaining steady internal chemical and physical conditions in the body by controlling blood pressure, acidity, hemoglobin levels, and regulating the electrolyte balance. Good kidney function is directly dependent on normal blood flow, making it an important renal biomarker. Acute kidney injury and impaired perfusion are in fact closely associated [2,3]. Furthermore, cortical perfusion has been found to be decreased in patients suffering from chronic kidney disease [4,5,6,7] and to vary depending on the disease stage [8]. Perfusion is also critical for renal allografts [9,10,11] and allows differentiation between allograft rejection and tubular necrosis [12].

Non-invasive perfusion quantification of the kidneys is thus important for diagnosis and monitoring of kidney diseases as well as the assessment of renal allograft function. Especially non-invasive imaging techniques such as arterial spin labeling (ASL) magnetic resonance imaging (MRI) are promising methods for these applications. ASL makes use of blood water protons as endogenous tracer to image perfusion quantitatively [13,14]. For this, blood water protons present in the arteries delivering blood to the organ of interest are magnetically labeled and after an inflow time, the labeled blood flows into the imaging region. A labeled image containing signal from the labeled blood and surrounding static tissue is then subtracted from a control image acquired without any labeling of blood to obtain a perfusion-weighted image used for quantification.

Clinical application of ASL MRI is still limited and standardization is lacking despite recent efforts [15,16,17]. Variability in evaluation of organ perfusion can be caused by various effects. The measurement itself involves a large number of degrees of freedom, including the choice of labeling scheme, readout scheme, scanner, as well as subject compliance and individual anatomy. In addition to these data acquisition-related sources of variability, differences in processing pipelines across centers further hinder useful comparison of organ perfusion quantification. Simulated data mimicking in vivo acquisitions are of great value as they do not include any variability stemming from acquisition and allow focusing solely on data processing.

The purpose of this work was to generate synthetic renal ASL data sets of the kidneys simulating in vivo acquisitions and to test our in-house developed processing pipeline using this synthetic data. Synthetic renal ASL data were generated using body models from the XCAT phantom and perfusion was added using the general kinetic model. Sequence and ASL parameters were set in accordance with the current consensus [17]. Our in-house developed processing pipeline was then evaluated in terms of registration, quantification, and segmentation using the synthetic data.

## 2. Materials and Methods

### 2.1. Synthetic ASL Data Sets

Synthetic MRI data were generated based on the anatomical structures provided by five models of the XCAT phantom (77, 80, 92, 93, and 108) [18]. XCAT voxel values were first converted into MRI magnitude values using the spin echo sequence equation:(1)$$S=\;\rho \;\left(1-{\;e}^{-\mathrm{TR}/T_{1}}\right){\;e}^{-\mathrm{TE}/T_{2}}.$$

Tissue specific parameters (relative proton density ρ, T1 and T2 relaxation times at field strengths of 3 T) were taken from the literature [19,20,21]. Coronal-oblique slice position (rotation angle of 12°), voxel dimension of 3 mm × 3 mm × 5 mm repetition time TR = 5000 ms, and echo time TE = 23 ms were chosen to match recommendations for in vivo acquisitions [17]. Noise was modeled as an additive white Gaussian noise to produce signal-to-noise ratio similar to in vivo acquisitions. Respiratory motion during free-breathing acquisition was simulated by generating 100 synthetic MR images at equally spaced time points around the exhalation part of the breathing cycle. ASL data sets with a total of 51 images (one M0 and 25 control–label pairs) were then generated by randomly selecting 51 images from the available 100 time points. Background suppression used for control and labeled images was modeled by reducing the signal intensity of all control and labeled images to 20% of the signal of the M0 image [22]. Assuming the apparent longitudinal relaxation time of tissue equals the longitudinal relaxation time of arterial blood, the general kinetic model [23] was used to create both pulsed ASL (PASL) and pseudo-continuous ASL (PCASL) data sets. To reproduce healthy kidneys, a perfusion ratio of five was assumed between cortex and medulla [24], with cortical perfusion set to 250 mL/100 g/min and medullary perfusion set to 50 mL/100 g/min. Arterial transit times of 1123 ms and 1141 ms were assumed for the medulla and cortex, respectively [25]. In addition to data sets with healthy perfusion, a PCASL data set with abnormal perfusion in the right kidney (cortical perfusion of 100 mL/100 g/min and medullary perfusion of 20 mL/100 g/min) was also generated using the body model 92. For all PASL data sets, the inversion time TI and the labeling duration TI_1_ were set to 1800 ms and 1200 ms, respectively. For all PCASL data sets, the post-labeling delay PLD was set to 1200 ms and the labeling duration τ was set to 1600 ms. The synthetic ASL data sets presented here are available for download under https://doi.org/10.11588/data/QAHWSF (accessed on 28 July 2022 ).

### 2.2. ASL Processing Pipeline

The data analysis pipeline for renal ASL data is an in-house developed MATLAB script (Version 2020a, MathWorks, Natick, MA, USA), which includes registration, quantification, and segmentation steps. The registration is performed using the open-source elastix toolbox [26,27] and follows a groupwise strategy in which all ASL images (M0, all control and all labeled images) are registered to a mean image. It is a multi-resolution registration (isotropic Gaussian kernels 10, 8, 2, 4, 2, 1) with non-rigid transform, 3d-order B-spline interpolator, adaptive stochastic gradient descent optimizer [28], and principal-component-analysis-based metric (PCAMetric2) [29]. Perfusion quantification follows the current consensus for renal ASL [17]:(2)$$\mathrm{rbf}\;\left[\mathrm{mL}/100g/\min\right]=\frac{6000\;{*\;\lambda *\;\Delta M*e}^{-\mathrm{TI}/T_{1}}}{2*\alpha \;*\;0.93^{2}*M0\;{*\;\mathrm{TI}}_{1}}$$
for PASL data, and
(3)$$\mathrm{rbf}\;\left[\mathrm{mL}/100g/\min\right]=\frac{6000\;{*\;\lambda *\;\Delta M*e}^{-\mathrm{PLD}/T_{1}}}{2*\alpha \;*\;0.93^{2}*M0\;{*\;T}_{1}*\left(1-{\;e}^{-\tau /T_{1}}\right)}$$
for PCASL data.

According to the consensus [17], we assumed a T_1_ of blood of 1650 ms at 3 T, a blood-tissue partition coefficient λ of 0.9 mL/g, and a labeling efficiency α of 0.95 and 0.85 for PASL and PCASL, respectively, which is corrected by a factor of 0.93^2^ to account for two background suppression pulses. For the segmentation, whole kidneys are first segmented manually on the M0 image. A k-means clustering algorithm is then used to automatically segment cortex and medulla. The clustering is applied on the quantified perfusion map masked by the manually drawn whole kidney masks for left and right kidney separately. To further improve the accuracy of the medulla segmentation mask, an additional erosion step is performed. For this, the manually drawn whole kidney masks were eroded using [0 0 0 1 1; 0 0 1 0 0; 0 1 1 0 0; 1 1 0 0 0] and [1 1 0 0 0; 0 0 1 0 0; 0 0 1 1 0; 0 0 0 1 1] as structuring elements for left and right kidney, respectively. The medulla mask produced by the clustering algorithm is then multiplied with the eroded whole kidney mask to remove any voxels on the outer edge of the kidneys. A flowchart of the processing pipeline can be found in Appendix A.

### 2.3. Pipeline Evaluation

The quality of registration was assessed qualitatively using line profiles across the time dimension of the ASL data sets as well as quantitatively with mean structural similarity index measures (MSSIMs) [30] calculated for all possible image pairs of a data set. The quantification step was evaluated by comparing the mean cortical and medullary perfusion obtained from the analysis to the values assumed to generate the data sets. The segmentation results obtained from the processing pipeline were compared to the segmentation masks provided by the XCAT phantom. For this, the organ masks from the XCAT phantom were first transformed with the registration matrix used to correct for the respiratory motion and then rebinarized. Whole kidney, cortex, and medulla binary masks were finally compared using the Dice index [31].

## 3. Results

### 3.1. Synthetic ASL Data Sets

The M0, first control, and first labeled images of synthetic single-slice PASL and PCASL data sets are shown in Figure 1.

### 3.2. Registration

Horizontal and vertical line profiles for both right and left kidneys of a synthetic ASL data set are shown in Figure 2a. Some movement is visible between subsequent ASL images in both horizontal and vertical direction before registration. The movement is noticeably reduced after the registration as indicated by the smoother line profiles and the less abrupt changes in signal intensity. Quantitative evaluation of the registration for all available healthy data sets is shown in Figure 2b in terms of MSSIM distributions before and after registration. For each model, the MSSIMs increase after registration, indicating that the resemblance between all images compared increases with the registration. Before registration, mean and standard deviation of MSSIMs across all healthy data sets were 0.4 +/− 0.2 and 0.4 +/− 0.1 for PASL and PCASL, respectively. After registration, mean and standard deviation of MSSIMs across all healthy data sets were 0.5 +/− 0.2 and 0.5 +/− 0.1 for PASL and PCASL, respectively. The comparison includes the comparison of identical images, which explains the MSSIM outliers equal to 1 in each distribution. The lowest MSSIMs were obtained for comparisons between M0 and all control or labeled images as M0 differs from all other ASL images in terms of intensity due to the assumed background suppression. As expected, the MSSIM values differed little between PASL and PCASL data sets. Across all models, no difference in MSSIM distributions could be observed between left and right kidney (0.4 +/− 0.2 before registration and 0.5 +/− 0.1 after registration for both).

### 3.3. Quantification

Perfusion maps obtained for the healthy and abnormal PCASL data sets from model 92 are shown in Figure 3a. The decreased perfusion in the right kidney of the abnormal data set is well distinguishable from the normal perfusion. The distributions of perfusion values in each data set for whole kidney, cortex, and medulla are shown in Figure 3b. A clear difference between both data sets is visible for the right kidney with mean perfusion and standard deviation of 220 +/− 40 mL/100 g/min and 90 +/− 30 mL/100 g/min for the cortex and 50+/− 50 mL/100 g/min and 10 +/− 30 mL/100 g/min for the medulla for healthy and abnormal data sets, respectively. At the same time, both data sets presente very similar perfusion distributions for the left kidney with mean and standard deviation of 210 +/− 50 mL/100 g/min for the cortex and 30 +/− 50 mL/100 g/min for the medulla. This indicates a good reproducibility of the generation of synthetic ASL data as well as the quantification step of the processing pipeline.

Looking at all the data sets with healthy perfusion (Figure 4), a narrow interquartile range of mean perfusion can be observed for whole kidney, cortex, and medulla for both left and right kidneys and for both labeling strategies. For the whole kidney, mean and standard deviation of renal perfusion across all healthy datasets were 150 +/− 40 mL/100 g/min and 130 +/− 10 mL/100 g/min for PASL and PCASL, respectively. For the cortex, mean and standard deviation of renal perfusion across all healthy datasets were 240 +/− 40 mL/100 g/min and 210 +/− 20 mL/100 g/min for PASL and PCASL, respectively. For the medulla, mean and standard deviation of renal perfusion across all healthy datasets were 50 +/− 20 mL/100 g/min and 49 +/− 8 mL/100 g/min for PASL and PCASL, respectively. This corresponds well to the perfusion values originally assumed to generate the synthetic data (250 mL/100 g/min and 50 mL/100 g/min for the cortex and medulla, respectively).

### 3.4. Segmentation

A comparison of segmentation masks obtained for one of the data sets is shown in Figure 5a. The segmentation masks for whole kidney, cortex, and medulla obtained from the processing pipeline match the XCAT segmentation masks well. Dice index distributions across all models for PASL and PCASL data sets can be found in Figure 5b. Very good agreement between segmentations was found for whole kidney and cortex, with dice indices ranging from 0.80 to 0.93 and from 0.79 to 0.89, respectively. Differences between medulla masks were larger but still reasonably good, with Dice indices ranging from 0.64 to 0.84.

## 4. Discussion

Synthetic renal ASL data sets simulating in-vivo acquisitions were generated using body models from the XCAT phantom, the general kinetic model and literature values for tissue properties. Sequence and ASL parameters were set in accordance with the current consensus [17]. The synthetic data were then used to evaluate our in-house developed renal ASL processing pipeline consisting of registration, quantification, and segmentation steps.

The generation of synthetic renal ASL data presented in this work allows flexible choice of parameters and benefits from the multitude of available XCAT body models presenting different organ sizes and shapes. An additional style transfer to further increase the resemblance between synthetic and real ASL data would be beneficial. This could be conducted using CycleGAN networks as previously demonstrated [32]. The five models included in this work were selected at random from the available adult models. Future work should also involve the expansion of the synthetic data sets to additional models as well as include additional pathological data sets (e.g., representing different stages of chronic kidney disease). In 2019, Antolak et al. introduced a simple ASL digital reference object to compare software packages [33]. It consists of a square block of voxels with a range of perfusion values (10–210 mL/100 g/min), considering only PCASL labeling and including noise. This simple approach does not resemble real-life cases and thus allows only limited evaluation or comparison of processing software. More recently a brain ASL digital reference object was introduced [34] and used to set up a challenge for comparison of ASL processing options employed in research and clinical settings [35]. The authors also looked into generating synthetic ASL data of the kidneys using the same framework [36]. Their approach used cortex and medulla segmented from an in vivo MR scan of a single healthy volunteer and a perfusion of 215 mL/100 g/min and 81 mL/100 g/min were assigned to cortex and medulla, respectively. This approach requires actual MRI acquisitions, which can be challenging to obtain due to scanner availability and potential artifacts.

Good registration to correct for respiratory motion is crucial for accurate perfusion quantification based on ASL MRI data as it requires the subtraction of labeled images from control images. A mismatch of anatomical structures between any of the images used for calculation of the perfusion map would result in erroneous perfusion values. The registration strategy employed in our in-house developed processing pipeline performs well on the synthetic data. A separate registration of left and right kidney was chosen as each organ may experience a different motion during the respiratory cycle. As a coronal oblique slice positioning was used, the respiratory motion, which is mainly along the craniocaudal direction, will result in a displacement of the kidneys not only within the plane of imaging but also out of it. This can result in varying organ shapes across the ASL time series and motivated the choice of a non-rigid registration strategy. Finding the best suited registration strategy and parameters is difficult and some adaption might be necessary when analyzing data from different cohorts. The performance of groupwise registration applied to ASL might be hindered by the difference of signal intensity between M0, control, and labeled images. Another registration strategy which has been shown to yield good results for motion correction in ASL of the myocardium consists of a combination of groupwise registration of control and of labeled images followed by pairwise registration of mean control image and mean labeled image to the M0 image [37]. Variational frameworks for non-rigid registration have been shown to yield good results for dynamic contrast enhanced (DCE) MR images of the kidneys [38] and could be evaluated for renal ASL. Deep learning based registration is also an interesting alternative as it has already demonstrated great potential [39]. However, the success and transferability of this approach is limited by the amount of data available for training. The framework for generation of synthetic ASL data presented here can be used to complement in vivo acquired data for network training purposes, as previously suggested for other imaging applications [32].

The quantification model employed here was a single compartment model and assumes the same longitudinal relaxation for tissue and blood. While this corresponds to the current recommendations [17], it contains a number of assumptions simplifying the complexity of blood and nutrient delivery to an organ. The framework for generation of synthetic ASL data presented here can easily be modified to include perfusion-weighting in the images based on a different and more complex model. In addition, it offers a large flexibility to modify other assumed tissue properties to simulate diseases involving impaired renal perfusion as well as altered tissue relaxation times and arterial transit times.

The segmentation procedure employed in our in-house developed processing pipeline is a compromise between time-intensive manual segmentation and accuracy of automated segmentation. Automated whole kidney segmentation is complicated, especially for the left kidney due to the closeness of the spleen, which presents similar intensities as the kidneys in the M0 image used for segmentation. Automated segmentation of cortex and medulla works well on the quantified perfusion map as perfusion differs greatly between cortex and medulla in healthy kidneys. However, this approach will not yield good results in cases where cortical and medullary perfusion are similar and further improvement is warranted for application in clinical cohorts. Kidney segmentation based on k-means clustering has previously been used on entire MR images with three clusters (cortex, medulla, background) [40] instead of running it on each kidney separately and thus using only two clusters (cortex and medulla) as done in this work. In our ASL processing pipeline, the image is already separated in two for separate registration of left and right kidney. A segmentation performed separately on each kidney is thus more straightforward. Another automated segmentation for DCE MR images yielding good results uses wavelet-based clustering [41]. However, it is not directly applicable to ASL images as it makes use of the DCE-specific changes in signal intensity during contrast agent uptake, which reflects kidney functionality. An automated segmentation of the kidneys in twelve concentric objects (TLCO) has been introduced for evaluation of renal oxygenation [42]. However, this method does not allow separation of cortical and medullary compartments of the kidneys and is thus less appealing for assessment of renal perfusion. Deep learning has also successfully been applied for kidney segmentation for both healthy and diseased kidneys [43,44]. In most cases, segmentation is performed on anatomical T1-weighted or T2-weighted data or even on quantified relaxometry maps [44,45]. However, this requires an additional step of registration of the anatomical and ASL data, which is not always easily achievable.

## 5. Conclusions

Synthetic renal ASL data were generated. They simulated in vivo acquired data based on current acquisition recommendations. The data were used to evaluate our in-house developed processing pipeline, which yielded good results for the registration, quantification, and segmentation steps. The synthetic data are well suited for evaluation of processing pipelines and could be used to compare pipelines across centers in a next step.

## Figures and Tables

**Figure 1 diagnostics-12-01854-f001:**
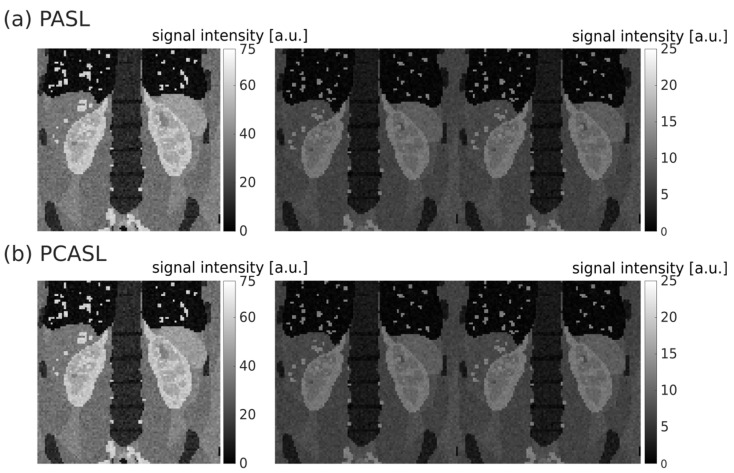
Exemplary PASL (**a**) and PCASL (**b**) single-slice data sets (model 77). Left column: M0 images; middle column: first control images; right column: first labeled images. The signal intensity of control and labeled images is lower than that of the M0 due to background suppression.

**Figure 2 diagnostics-12-01854-f002:**
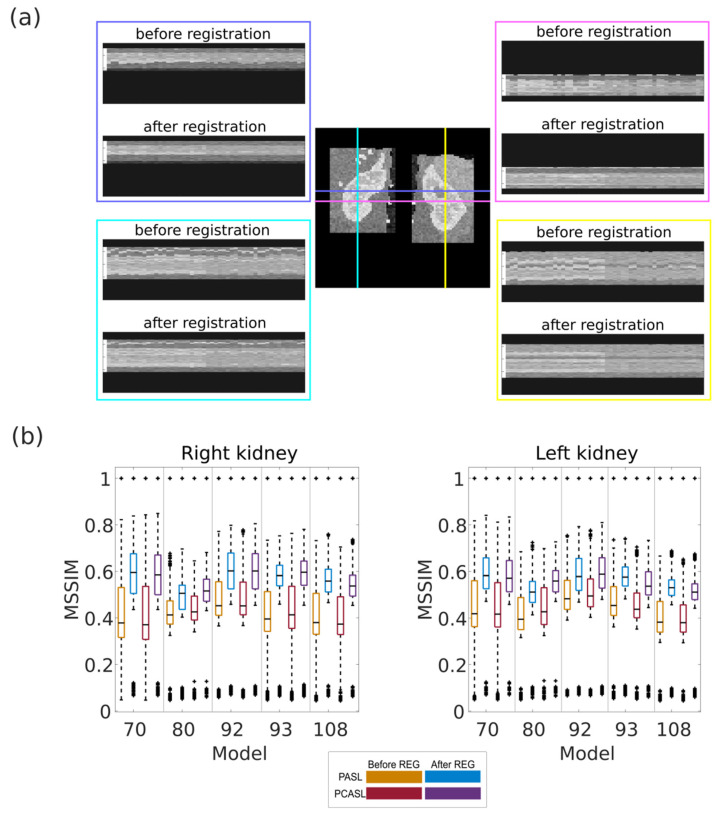
(**a**) Line profiles across all ASL images before and after registration (model 93, PASL). (**b**) MSSIMs calculated before and after registration for all possible image pairs within each healthy data set. The black line in each boxplot shows the median of the distribution of mean perfusion across all models. The top and bottom part of each boxplot represent the 25th and 75th percentiles. The boxplots’ whiskers extend to the highest and lowest data points not considered as outliers. Outliers are shown a black ‘+’.

**Figure 3 diagnostics-12-01854-f003:**
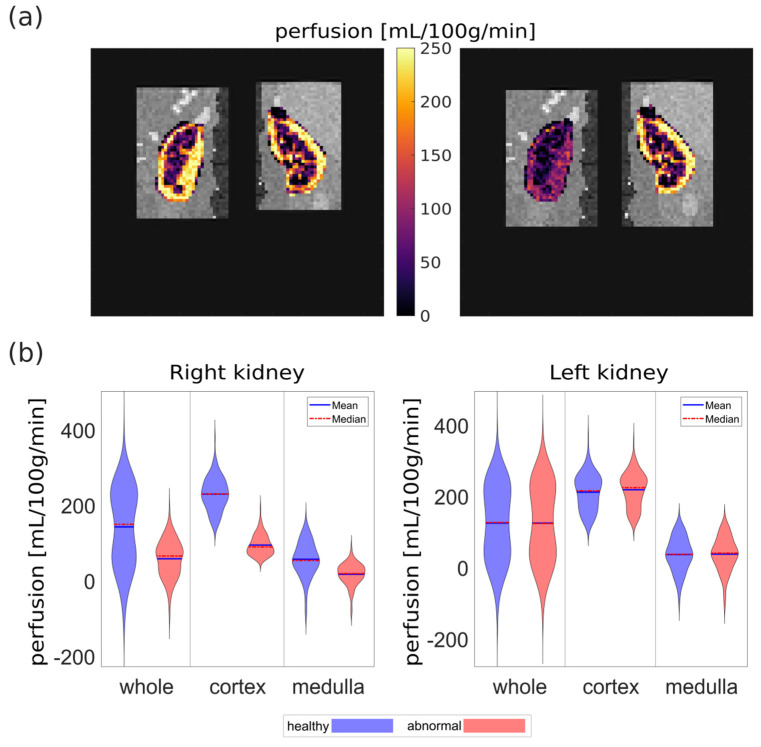
(**a**) M0 image cropped to rectangles used for separate left and right registration with overlaid perfusion map obtained for the healthy (**left**) and abnormal (**right**) PCASL data sets of model 92. (**b**) Distributions of perfusion values for both data sets.

**Figure 4 diagnostics-12-01854-f004:**
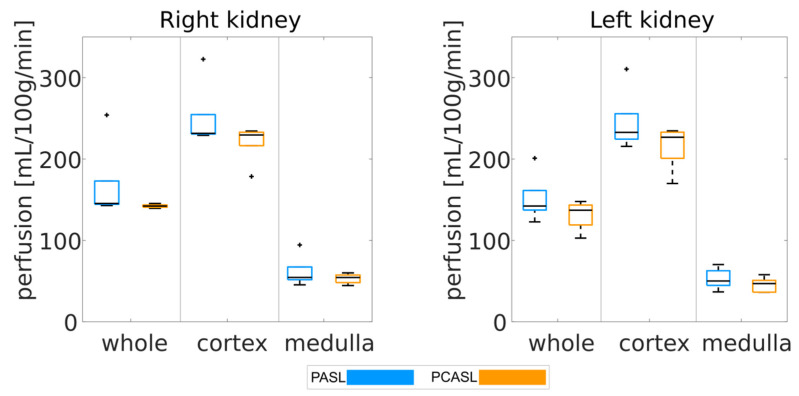
Mean renal perfusion averaged over the whole kidney, the cortex, and the medulla for all considered models with healthy perfusion. The black line in each boxplot shows the median of the distribution of mean perfusion across all models. The top and bottom part of each boxplot represent the 25th and 75th percentiles. The boxplots’ whiskers extend to the highest and lowest data points not considered as outliers. Outliers are shown a black ‘+’.

**Figure 5 diagnostics-12-01854-f005:**
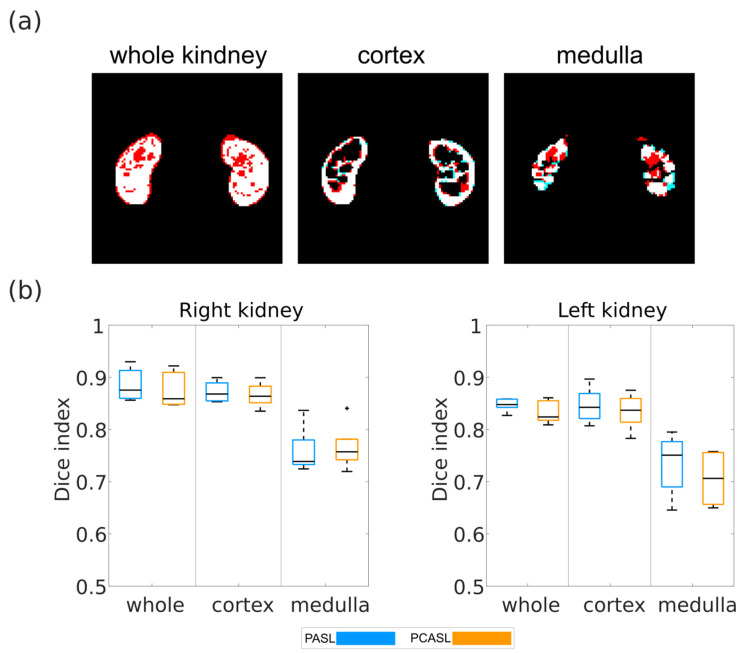
(**a**) Comparison of segmentation masks of the PCASL data set of model 108 obtained from the processing pipeline (red) and the XCAT phantom (cyan). White areas correspond to pixels present in both masks being compared. (**b**) Boxplots of Dice indices calculated for the five models for the PASL and PCAL data sets for whole kidney, cortex, and medulla. The black line in each boxplot shows the median of the distribution of mean perfusion across all models. The top and bottom part of each boxplot represent the 25th and 75th percentiles. The boxplots’ whiskers extend to the highest and lowest data points not considered as outliers. Outliers are shown a black ‘+’.

## Data Availability

The synthetic data presented here is available for download under https://doi.org/10.11588/data/QAHWSF (accessed on 28 July 2022).

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
