# Peer review of "Synthetic Arterial Spin Labeling MRI of the Kidneys for Evaluation of Data Processing Pipeline"

_diagnostics, 2022, doi:10.3390/diagnostics12081854_

Round 1
Reviewer 1 Report
OK, in principle nice work and well presented.
For me a little disappointing that the core work is mainly simulations, for me the excitement is then a little far away. For me it is hard to really appreciate the results without patient cases being treated.
Author Response
This work does not include any patient cases as this was not the focus of this project. This work aims to facilitate standardization of renal ASL for clinical applications. At the moment, ASL is used for renal perfusion quantification in patients at some centers but the lack of ground truth perfusion knowledge and differences in processing pipelines between centers impede wide-spread transition to the clinical routine. Standardization is helped by simulations, which intrinsically provide a ground truth for the scenarios simulated and can support the understanding of sources of variability between centers. Simulated data mimicking in-vivo acquisitions is of great value as it does not include any variability stemming from acquisition and allows focusing solely on differences arising from data processing.
Reviewer 2 Report
Overall, the purpose and results of this paper are clear, with a few minor issues:
1. If the subject of the paper emphasizes that this is an improved "pipeline" to enhance its ability to segment kidney tissue and facilitate its effect on perfusion prediction, please provide a flowchart of the pipeline. After I read it, it seems that "registration" is the most important and new step among them.
2. This "registration" step seems to be a voxel alignment technique, so that the blurred voxels can be correctly arranged and the correct tissue segmentation results can be obtained. But this step is placed in the results paragraph. Why wasn't he put in the method?
3. The article does not seem to particularly emphasize the reason why this "registration" step was added, and the comparison with previous research methods (there is a comparison of the results, I mean the necessity of this step being added is not very clear)
Author Response
1.
The emphasis is not on an improved processing pipeline but rather on the generation of synthetic ASL MRI data, which can be used to evaluate and compare processing pipelines. In addition to presenting the data, we evaluate our in-house developed pipeline with this synthetic data to demonstrate its usability and provide guidance for other centers interested in evaluating processing pipelines for renal ASL data. The registration is indeed an important step but our approach is not new.
2.
The registration used here is a voxel-by-voxel deformable transformation applied to all ASL images to ensure identical anatomical structures are at the same position in each image. Details about the registration are included in the methods part:
The registration is performed using the open-source elastix toolbox [26,27] and follows a groupwise strategy in which all ASL images (M0, all control and all labeled images) are registered to a mean image. It is a multi-resolution registration (isotropic Gaussian kernels 10, 8, 2, 4, 2, 1) with non-rigid transform, 3d-order B-spline interpolator, adaptive stochastic gradient descent optimizer [28] and principal-component-analysis-based metric (PCAMetric2) [29].
3.
The registration step is crucial for most abdominal imaging involving image acquisition over time as these images will contain some organ movement caused by the subject’s respiration. In some cases, image acquisition in breath-hold or using respiratory navigation can be valid alternatives to minimize differences in organ position between acquired images. However, in our case, the processing pipeline was developed for free-breathing renal ASL acquisitions, which thus include respiratory motion and require registration. And the synthetic data presented also includes respiratory motion. As already stated in the discussion: Good registration to correct for respiratory motion is crucial for accurate perfusion quantification based on ASL MRI data as it requires the subtraction of labeled images from control images.
To clarify this further, a sentence was added in the discussion:
A mismatch of anatomical structures between any of the images used for calculation of the perfusion map would result in erroneous perfusion values.